# Predictors of Repeat Medical Emergency Team Activation in Deteriorating Ward Patients: A Retrospective Cohort Study

**DOI:** 10.3390/jcm11061736

**Published:** 2022-03-21

**Authors:** Ju-Ry Lee, Youn-Kyung Jung, Sang-Bum Hong, Jin Won Huh

**Affiliations:** 1Department of Nursing, Geoje University, 91, Majeon 1-gil, Geoje 53325, Korea; pr20014@koje.ac.kr; 2Medical Emergency Team, Asan Medical Center, 88, Olympic-ro 43-gil, Songpa-gu, Seoul 05505, Korea; whitej11@hanmail.net; 3Department of Pulmonary and Critical Care Medicine, Asan Medical Center, University of Ulsan College of Medicine, 88, Olympic-ro 43-gil, Songpa-gu, Seoul 05505, Korea; sbhong@amc.seoul.kr

**Keywords:** hospital rapid response team, risk factors, mortality, critical care

## Abstract

Recurrent clinical deterioration and repeat medical emergency team (MET) activation are common and associated with high in-hospital mortality. This study assessed the predictors for repeat MET activation in deteriorating patients admitted to a general ward. We retrospectively analyzed the data of 5512 consecutive deteriorating hospitalized adult patients who required MET activation in the general ward. The patients were divided into two groups according to repeat MET activation. Multivariate logistic regression analyses were used to identify the predictors for repeat MET activation. Hematological malignancies (odds ratio, 2.07; 95% confidence interval, 1.54–2.79) and chronic lung disease (1.49; 1.07–2.06) were associated with a high risk of repeat MET activation. Among the causes for MET activation, respiratory distress (1.76; 1.19–2.60) increased the risk of repeat MET activation. A low oxygen saturation-to-fraction of inspired oxygen ratio (0.97; 0.95–0.98), high-flow nasal cannula oxygenation (4.52; 3.56–5.74), airway suctioning (4.63; 3.59–5.98), noninvasive mechanical ventilation (1.52; 1.07–2.68), and vasopressor support (1.76; 1.22–2.54) at first MET activation increased the risk of repeat MET activation. The risk factors identified in this study may be useful to identify patients at risk of repeat MET activation at the first MET activation. This would allow the classification of high-risk patients and the application of aggressive interventions to improve outcomes.

## 1. Introduction

The rapid response system (RRS) is used to identify clinically deteriorating patients and is used across several areas in hospitals [1]. Patients considered at risk by the medical emergency team (MET) may be admitted to the intensive care unit (ICU) or to the wards, depending on their clinical decision [2,3]. MET activation has been shown to reduce unexpected ICU admissions, cardiac arrest, and hospital mortality [4,5,6]. An increase in the utilization of MET services is associated with good clinical outcomes [7]. However, repeat MET activation in at-risk or deteriorating patients is associated with a high risk of ICU admission, prolonged hospitalization, and high in-hospital mortality [2,8]. Stelfox et al. [2] reported that repeat MET activation was more common in patients with respiratory diseases requiring airway management, such as suctioning, intubation, and mechanical ventilation. Meanwhile, Calzavacca et al. [8] reported a higher probability of multiple MET activations in patients undergoing surgery, those with gastrointestinal conditions at admission, and those with arrhythmia. The purpose of this study was to analyze the clinical characteristics of patients with repeat MET activation and to identify the factors at the first MET activation that predict repeat MET activation in patients admitted to general wards.

## 2. Materials and Methods

### 2.1. Study Cohort

To analyze the clinical characteristics of patients with repeat MET activation in the general ward, this retrospective cohort study evaluated the data of consecutive adult patients with a sudden clinical deterioration that triggered an initial MET activation between 1 January 2012 and 31 December 2016 at Asan Medical Center. Repeat MET activation usually occurs within 48 h of a previous MET activation [8]. Therefore, we included all patients with repeat MET activation for whom MET was reactivated within 48 h of the first activation. The exclusion criteria were as follows: (1) the implementation of a “do-not-resuscitate” (DNR) order before MET activation; (2) the MET was contacted to perform cardiopulmonary resuscitation; (3) MET activation was performed for educational or procedural purposes, such as insertion of a central line or examination of a patient using a portable ultrasound machine for patients without deterioration; (4) an interval of >48 h between the first and second MET activations; (5) transfer to the ICU after the first MET activation; (6) missing data needed for calculation of the Sequential Organ Failure Assessment (SOFA) score and Modified Early Warning Score (MEWS).

The protocols of this study were approved by the Institutional Review Board of Asan Medical Center (2017-0601), which waived the need for informed consent due to the retrospective nature of the study. The data of all study patients were deidentified before analyses for confidentiality.

### 2.2. The Medical Emergency Team

The MET system was introduced in our hospital in March 2008. The MET consists of three intensivists, four ICU fellows, two internal medicine residents, and nine clinical nurse specialists (CNSs) with experience in critical care. Every duty is handled by a team of at least one intensivist or fellow, one resident, and two CNSs. The MET can be activated 24 h a day when (1) nurses or doctors call the MET for aid, (2) the MET CNS identifies a high-risk patient admitted to a general ward using the electronic medical record-based automatic screening system, which predefines patients’ vital signs or laboratory value thresholds, or (3) a code blue is announced for cardiopulmonary arrest as published previously [9]. In our hospital, the MET makes ICU disposition decisions for deteriorating patients admitted to a general ward due to the limited ICU beds.

### 2.3. Data Collection

Data were collected prospectively using case forms and were analyzed retrospectively. The case forms were completed by the MET CNS, who participated in the MET activation and was trained in data collection methods to minimize bias in data collection. Variables included demographic characteristics, comorbidity, reasons for MET activation, vital signs, diagnostic and therapeutic interventions, patient disposition, laboratory data, the ratio of arterial oxygen saturation to the fraction of inspired oxygen concentration (SpO_2_/FIO_2_ (SF ratio)), the SOFA score, the MEWS, and the result of MET activation (stay in the general ward or transfer to ICU). The MEWS consists of variables, including systolic blood pressure, pulse rate, respiratory rate, body temperature, and mental status [10] at the time of MET activation. The SF ratio is a useful marker for predicting the condition and prognosis of deteriorating patients admitted to a general ward without arterial cannulation [11,12]. Therefore, we collected data on the SF ratio derived noninvasively using pulse oximetry [11,12]. In addition, hospital discharge data and the date of death of patients with MET activation were collected every 6 months. Variables that were not documented at the time of MET activation were treated as missing values. The patients were divided into two groups according to repeat MET activation status. The outcomes were in-hospital mortality and mortality within 28 days after MET activation.

### 2.4. Statistical Analysis

Between-group differences were assessed using the chi-square test. Fisher’s exact test was used to compare categorical variables, while the Mann–Whitney U test was used to compare continuous variables. The results are expressed as number (%) for categorical variables and as median with interquartile range (IQR) for continuous variables, as most continuous variables were not normally distributed. Univariate and multivariate logistic regression analyses were used to identify factors associated with recurrent deterioration and repeat MET activation. Variables that were significant in the univariate analyses were adjusted and included in the multivariate analyses to investigate causality. The results of these analyses are reported as odds ratios (ORs) with 95% confidence intervals (CIs). Two-sided *p*-values below 0.05 were considered statistically significant. All statistical analyses were performed using IBM SPSS Statistics version 21 (IBM Corp., Armonk, NY, USA).

## 3. Results

### 3.1. Patient Characteristics

Of 9468 patients who were identified, we excluded 3956 patients: patients with a DNR order before MET activation (*n* = 487, 5.1%), procedure/education without clinical deterioration (*n* = 788, 8.3%), cardiac arrest (*n* = 570, 6.0%), an interval of >48 h between the first and second MET activations (*n* = 52, 0.5%), transfer to ICU after the first MET activation (*n* = 1906, 20.1%), and activation with insufficient data for calculating the SOFA score and MEWS (*n* = 153, 1.6%). Finally, 5512 patients were included in the analysis (Figure 1). Fifteen percent of the patients (*n* = 840) experienced repeat MET activation during the same hospital stay.

The most common time intervals between the first and second MET activations were 24 h (*n* = 601, 67.4%) and 48 h (*n* = 239, 26.8%) (Figure 2). Repeat MET activation within 24 h of the first activation was more common in patients with the same indication during both activations (306/601, 50.1%).

The patients’ baseline characteristics are shown in Table 1. The median age was 64 (IQR, 54–73) years; 61% of the patients were men, and 85% were admitted to the medical ward. The number of patients admitted to the medical ward was higher in the repeat activation group than in the single activation group. There were no significant differences in age and sex between the two groups. The most common comorbidities were solid tumors and chronic heart disease. Hematological malignancies and chronic lung disease were more common in the repeat activation group than in the single activation group. Respiratory distress was the more common reason for repeat MET activation in the repeat activation group than in the single activation group. Meanwhile, the most common reason in the single activation group was sepsis/septic shock, hypovolemic shock, and provider concern. The repeat activation group had a significantly higher heart rate, respiratory rate, FiO_2_, MEWS, and SOFA score, as well as a significantly lower SF ratio, than the single activation group.

### 3.2. Interventions during MET Activation

We compared the interventions during MET activation according to repeat MET activation within the same admission. Airway suctioning (28.8% vs. 4.6%, *p* < 0.001), noninvasive mechanical ventilation (4.0% vs. 2.1%, *p* = 0.001), high-flow nasal cannula (HFNC) oxygenation (32.6% vs. 4.8%, *p* < 0.001), peripheral insertion of intravenous catheter (11.2% vs. 8.5%, *p* = 0.015), and vasopressor support (10.6% vs. 4.8%, *p* < 0.001) were more common in the repeat activation group than in the single activation group (Table 2).

### 3.3. Outcomes

Fifty-two percent (437/840) of the patients in the repeat activation group were admitted to the ICU. The likelihood of death within 28 days (OR, 1.34; 95% CI, 1.03–1.74; *p* < 0.028) or during hospitalization (OR, 2.05; 95% CI, 1.61–2.61; *p* < 0.001) was significantly higher in the repeat activation group. This was same for patients without any DNR order after MET activation (OR, 1.56; 95% CI, 1.19–2.04; *p* < 0.001 and OR, 1.75; 95% CI, 1.37–2.24; *p* < 0.001, respectively) (Table 3). 

### 3.4. Risk Factors of Repeat MET Activation

Seventeen variables were identified in the univariate logistic regression analysis and included into the multivariate logistic regression analysis (Table 4). Hematological malignancies (OR, 2.07; 95% CI, 1.54–2.79; *p* < 0.001) and chronic lung disease (OR, 1.49; 95% CI, 1.07–2.06; *p* = 0.017) were associated with a high risk of repeat MET activation. Among the causes for MET activation, respiratory distress (OR, 1.76; 95% CI, 1.19–2.60; *p* < 0.001) increased the risk of repeat MET activation. Patients with a low SF ratio (OR, 0.97; 95% CI, 0.95–0.98; *p* < 0.001), HFNC oxygenation (OR 4.52; 95% CI, 3.56–5.74; *p* < 0.001), airway suctioning (OR, 4.63; 95% CI, 3.59–5.98; *p* < 0.001), noninvasive mechanical ventilation (OR, 1.52; 95% CI, 1.07–2.68; *p* < 0.001), and vasopressor support (OR, 1.76; 95% CI, 1.22–2.54; *p* = 0.002) at the time of MET activation had a high risk of repeat MET activation.

### 3.5. Risk Factors of Repeat MET Activation by Respiratory Causes

We performed a subgroup analysis of the data of patients with MET activation for a respiratory cause. We divided the patients into two groups according to repeat MET activation (Appendix A). Of the 2693 patients for whom MET was activated due to respiratory causes, MET activation was repeated in 531 (19.7%) patients. The median age was 66 (IQR, 54–74) years; 64.5% of the patients were men, and 88.7% were admitted to the medical ward. The number of medical patients admitted for hematological malignancy (*p* < 0.001) or chronic lung disease (*p* = 0.011) was higher in the repeat activation group than that in the single activation group. Furthermore, the repeat activation group had a significantly higher FiO_2_ (*p* < 0.001), lower SpO_2_ (*p* < 0.001), and lower SF ratio (*p* < 0.001) than the single activation group. Regarding interventions, the rates of HFNC oxygenation (*p* < 0.001) and airway suctioning (*p* < 0.001) were higher in the repeat activation group than those in the single activation group.

To identify the risk factors associated with repeat MET activation for a respiratory cause, we adjusted eight variables identified in the univariate logistic regression analysis and included them in the multivariate logistic regression analysis (Table 5). The results showed a higher risk of repeat MET activation for respiratory causes in patients with hematological malignancies (OR, 2.65; 95% CI, 1.94–3.62; *p* < 0.001), a low SF ratio (OR, 0.98; 95% CI, 0.98–0.99; *p* < 0.001) at the time of MET activation, HFNC oxygenation (OR, 2.86; 95% CI, 2.22–3.72; *p* < 0.001), and airway suctioning (OR, 4.98; 95% CI, 3.69–6.71; *p* < 0.001).

## 4. Discussion

We performed a retrospective cohort study to identify the predictors for repeat MET activation in deteriorating patients admitted to a general ward. We found that repeat MET activation is independently associated with hematological malignancies, chronic lung disease at admission, vasopressor support, and respiratory distress-related factors, such as a low SF ratio, HFNC oxygenation, noninvasive mechanical ventilation, and airway suctioning after the first MET activation. Furthermore, we found that repeat MET activation for at-risk or deteriorating patients is common and associated with high hospital mortality.

In our study, approximately 15% of deteriorating patients admitted to a general ward requiring MET activation experienced recurrent clinical deterioration and a repeat MET activation. This is similar to the rate of 17% reported in a study by Konard et al. [5]. Conversely, Stelfox et al. [13] and Calzavacca et al. [8] reported higher rates of 10.5% and 22.5%, respectively. The incidences of repeat MET activation in the previous studies [2,5,8] are broadly similar, and the slight differences could be explained by the patients’ characteristics, institutional characteristics, and MET practice patterns [8].

In our study, repeat MET activation within 24 h of the first activation was more common among patients with repeat MET activation for the same reason in the first activation (50.1%). Calzavacca et al. [8] reported that most repeat MET activations occurred within 48 h. Furthermore, previous studies [8,13] reported that most patients had the same physiological triggers for repeat MET activation. This suggests that, for most patients, the etiology of recurrent clinical deterioration is likely to be the etiology of deterioration of their original condition, wrong diagnosis, wrong treatment, or refractory disease. By analyzing the triggers of MET activations (e.g., respiratory rate), it may be possible to identify patients at risk of recurrent clinical deterioration (e.g., consistently increased respiratory rate), to identify the reason for rescue failure, and to intervene before the situation escalates to an emergency [14]. Repeat MET activation is analogous to ICU readmission in many ways; their reduction has been attempted for decades [15]. This suggests that repeat MET activation may be a valuable quality metric for RRS and that monitoring is likely a first step in developing strategies to further optimize MET performance.

In our study, repeat activation was associated with a higher risk of in-hospital mortality, and these results are consistent with those of previous studies [2,3,8]. Repeat MET activation is associated with statistically and clinically significant increases in healthcare service utilization and mortality [8,13]. This suggests that the consequences of repeat MET activation are sufficient to warrant efforts to identify patients at risk of recurrent clinical deterioration.

In our study, repeat MET activation was associated with hematologic malignancies, chronic lung disease at admission, vasopressor support, and factors related to respiratory distress, such as a low SF ratio, HFNC oxygenation, noninvasive mechanical ventilation, and airway suctioning after the first MET activation. Previous studies have reported that repeat MET activation is associated with chronic liver disease at admission and procedures performed after the initial MET activation, including airway suctioning and noninvasive mechanical ventilation [2,8,16]. To the best of our knowledge, this is the first report showing that repeat MET activation was associated with hematologic malignancies, chronic lung disease at admission, a low SF ratio, HFNC oxygenation, and vasopressor support at the time of first MET activation. These factors could provide a basis for identifying high-risk patients. Compared to previous studies [2,8,16], common reasons for repeat MET activation are factors associated with respiratory distress or respiratory care, although the underlying diseases might differ.

In our system, although patients with respiratory distress are hemodynamically stable, noninvasive mechanical ventilation or HFNC oxygenation is performed and monitored by respiratory therapists, preferentially in the ward. Simultaneously, the MET monitors and manages the patient until the patient’s condition is stabilized. In settings with limited resources, high-risk patients may be managed with the time-limited trial of these interventions, provided there are adequate resources for close monitoring. However, any deterioration should prompt transfer to a higher level of care. Our results show that adequate assessment and monitoring are important when these interventions are performed in the ward.

Notably, in our study, there were a significantly high number of patients with MET activation due to respiratory distress in the repeat activation group than in the single activation group. Furthermore, our study is the first to identify the risk factors for repeat MET activation for respiratory causes. Our results showed a higher risk of repeat MET activation for respiratory causes in patients with hematological malignancies, a low SF ratio at the time of MET activation, HFNC oxygenation, and airway suctioning. The condition of patients with clinical deterioration due to respiratory distress progressed slower than that of patients with deterioration due to septic shock or hypovolemic shock; these patients were managed initially with oxygen therapy. This means that it can be challenging to predict the prognosis of patients with respiratory distress; thus, a respiratory specialist might be needed in the MET. For these patients, mandatory rounds and early decision making concerning ICU admission may be helpful. The management of a deteriorating patient admitted to a general ward with repeat MET activation is clearly complex, as it is influenced by the patient, healthcare provider, and hospital’s characteristics [2,8,13]. This suggests that the development of validated MET guidelines may facilitate more accurate management of patients with clinical deterioration and MET activation by enabling standardized decision making regarding post-MET monitoring and the most appropriate location for patient care (e.g., current general ward vs. high-dependency unit vs. ICU) [2,17,18].

HFNC oxygenation is superior to conventional oxygen therapies to improve dyspnea and oxygenation [19]. HFNC is increasingly used in critically ill adult patients in various clinical settings [20]. In a randomized controlled study, HFNC use in patients with severe respiratory failure reduced the intubation rate and lowered mortality [21]. However, to the best of our knowledge, this is the first report identifying HFNC oxygenation during initial MET activation as a predictor of repeat MET activation. Kang et al. [22] reported that failure of HFNC oxygenation might cause a delay in intubation and worsen clinical outcomes in patients with respiratory failure. Therefore, HFNC oxygenation should be performed appropriately in deteriorating patients, and careful monitoring of such patients for a certain period is necessary.

Fernando et al. [3] reported that, for patients at risk of recurrent deterioration, there are few reliable tools that can be used prospectively. Because the oxygenation of most patients was monitored by pulse oximetry, we collected the value of the SF ratio: an index which has been validated as a noninvasive surrogate of the arterial oxygen pressure/fraction of inspired oxygen ratio (PaO_2_/FiO_2_) [12] and can predict the condition and prognosis of deteriorating patients admitted to a general ward without arterial cannulation [11,23]. Therefore, our results may help healthcare providers to identify patients at risk for recurrent deterioration early.

Our study has both strengths and limitations. Firstly, our study was performed in a single medical center. MET practice patterns and decision-making processes may differ in healthcare settings; therefore, the incidence of recurrent clinical deterioration and repeat MET activation may also differ with the institution. However, the observed incidence in our study was similar to that reported in the existing literature. Moreover, the challenge of patients with recurrent clinical deterioration and repeat MET activation is also common in other health organizations. Secondly, even with adjustment, undetected confounding variables (such as ICU resource availability) for repeat MET activation and patient outcomes may have been present, which may have led to incorrectly estimated associations. Thirdly, because invasive interventions such as noninvasive ventilation, HFNC oxygenation, vasopressor use, and frequent airway suctioning are routinely performed in the ward, the triage criteria for critically ill patients might differ in our hospital. Most healthcare facilities with limited ICU beds worldwide have similar practice patterns to ours [2,8,16].

Despite these limitations, the available data sources contained considerable data, including information on illness severity (e.g., the SOFA score and MEWS), which could help to further characterize patients at risk of repeat MET activation. The identified risk factors may be helpful to identify patients at risk of repeat MET activation and provide aggressive interventions to improve outcomes. Further studies in other healthcare settings are required to confirm our findings.

## 5. Conclusions

Repeat MET activation increases in-hospital mortality rates. It is independently associated with hematologic malignancies, chronic lung disease at admission, MET activation for respiratory causes, such as a low SF ratio, HFNC oxygenation, airway suctioning, vasopressor support, and noninvasive mechanical ventilation at the first MET activation. Thus, patients with these characteristics should be targeted for specific care and policies aimed at improving outcomes.

## Figures and Tables

**Figure 1 jcm-11-01736-f001:**
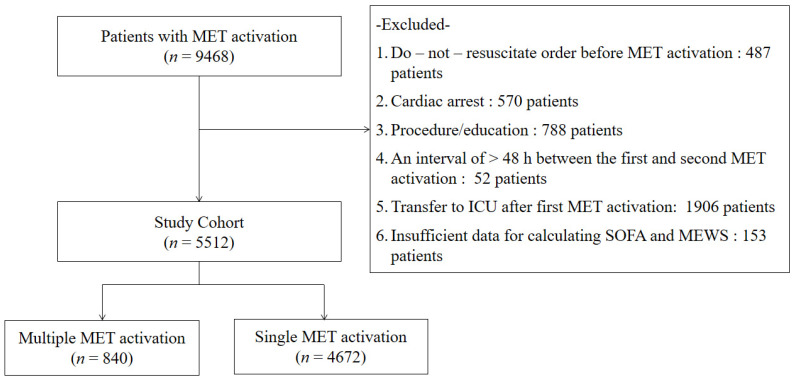
Flow diagram of patients. MET = medical emergency team; SOFA = Sequential Organ Failure Assessment; MEWS = Modified Early Warning Score.

**Figure 2 jcm-11-01736-f002:**
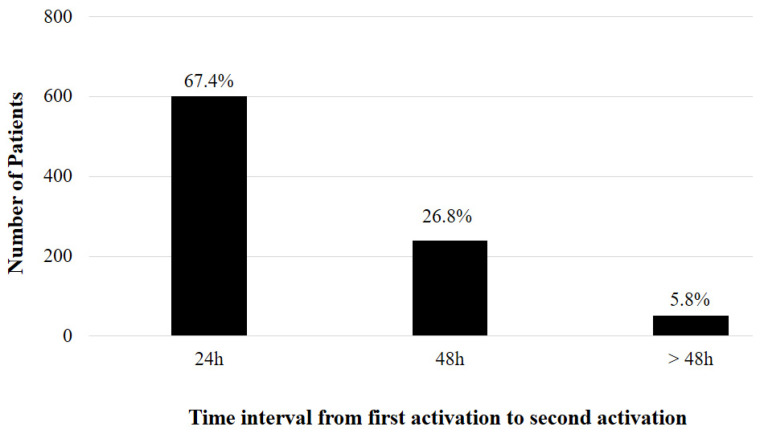
Prevalence of repeat medical emergency team after the first activation.

**Table 1 jcm-11-01736-t001:** Baseline characteristics of the patients at time of contact with medical emergency team.

Variables	Repeat MET Activation		*p*
	Yes (*n* = 840)	No (*n* = 4672)	
Age, years	64 (52–73)	64 (53–73)	0.739
Male gender	548 (65.2)	2943 (63.0)	0.432
Department			
Medicine	752 (89.5)	3931 (84.1)	<0.001
Surgery	88 (10.5)	741 (15.9)	
Comorbidities			
Solid tumor	267 (31.8)	1635 (35.0)	0.382
Hematological malignancies	210 (25.0)	435 (9.3)	<0.001
Chronic lung disease	141 (16.8)	500 (10.7)	<0.001
Chronic heart disease	383 (45.6)	1961 (42.0)	0.058
Chronic liver disease	95 (11.3)	696 (14.9)	0.053
Chronic renal disease	55 (6.5)	246 (5.3)	0.138
Cause for MET activation			
Respiratory distress	599 (71.3)	2162 (46.3)	<0.001
Sepsis/septic shock	116 (13.8)	938 (20.1)	<0.001
Hypovolemic shock	18 (2.1)	323 (6.9)	<0.001
Arrhythmia	11 (1.3)	107 (2.3)	0.071
Altered mental status	21 (2.5)	140 (3.0)	0.587
Metabolic acidosis	48 (5.7)	378 (8.1)	0.053
Provider worried	27 (3.2)	491 (10.5)	<0.001
Vital parameter at first MET activation			
Systolic blood pressure (mmHg)	119 (100–137)	110 (90–132)	<0.001
Heart rate (beats/min)	110 (94–128)	106 (88–122)	<0.001
Respiratory rate (breaths/min)	26 (22–32)	22 (20–28)	<0.001
Body temperature (°C)	36.9 (36.5–37.7)	36.8 (36.4–37.5)	<0.001
SpO_2_ (%)	95 (91–98)	96 (92–98)	0.059
FiO_2_ (%)	0.37 (0.25–0.45)	0.28 (0.21–0.45)	<0.001
SpO_2_/FiO_2_ ratio	213 (169–338)	350 (218–452)	<0.001
Modified Early Warning Score	5 (3–7)	4 (3–6)	<0.001
SOFA	7 (4–9)	4 (3–7)	<0.001

MET = medical emergency team; SOFA = Sequential Organ Failure Assessment. Data are expressed as median (interquartile range) or number (percentage).

**Table 2 jcm-11-01736-t002:** Intervention of care during first medical emergency team activation.

Variables	Repeat MET Activation		*p*
	Yes (*n* = 840)	No (*n* = 4672)	
Noninvasive mechanical ventilation	34 (4.0)	98 (2.1)	0.001
High-flow nasal cannula application	274 (32.6)	224 (4.8)	<0.001
Supply oxygen	161 (19.2)	980 (20.0)	0.273
Suctioning of the airway	242 (28.8)	217 (4.6)	<0.001
Arterial catheter inserted	36 (4.3)	265 (5.7)	0.117
Central IV catheter inserted	74 (8.8)	354 (7.6)	0.234
Peripheral IV catheter inserted	94 (11.2)	398 (8.5)	0.015
Fluid resuscitation	187 (22.3)	981 (20.1)	0.063
Transfusion	23 (2.7)	152 (3.3)	0.521
Vasopressor support	89 (10.6)	222 (4.8)	<0.001

MET = medical emergency team. Data are expressed as number (percentage).

**Table 3 jcm-11-01736-t003:** Clinical outcomes in patients according to repeat medical emergency team activation.

Outcomes	Repeat MET Activation		Unadjusted OR		Adjusted OR	
	Yes	No	OR (95% CI)	*p*	OR (95% CI)	*p*
28 day mortality overall	329/840 (37.2)	1111/4671 (23.8)	2.06 (1.77–2.41)	<0.001	1.34 (1.03–1.74)	0.028
28 day mortality (no DNR patients)	142/577 (24.6)	560/3422 (16.4)	1.67 (1.35–2.06)	<0.001	1.56 (1.19–2.04)	<0.001
In-hospital mortality overall	461/840 (54.9)	1406/4671 (30.1)	2.83 (2.43–3.28)	<0.001	2.05 (1.61–2.61)	<0.001
In-hospital mortality (no DNR patients)	193/577 (33.4)	729/3422 (21.3)	1.86 (1.53–2.25)	<0.001	1.75 (1.37–2.24)	<0.001

MET = medical emergency team; DNR = do not resuscitate. ORs and *p*-values were adjusted for age, sex, comorbidity (chronic lung disease, hematological malignancies, solid tumor), cause for MET activation, SOFA, MEWS, SpO_2_/FiO_2_ ratio, and intervention (noninvasive mechanical ventilation, high-flow nasal cannula apply, supply oxygen, suctioning of the airway, vasopressor support, or arterial catheter insertion) after MET activation.

**Table 4 jcm-11-01736-t004:** Univariate and multivariate logistic regression analyses for repeat medical emergency team activation.

Variables	Univariate		Multivariate	
	OR (95% CI)	*p*	OR (95% CI)	*p*
Age	1.00 (0.99–1.00)	0.484	1.00 (0.99–1.00)	0.196
Male (reference: female)	1.01 (0.88–1.35)	0.437	1.11 (0.88–1.40)	0.365
Medical department	1.61 (1.27–2.04)	<0.001	1.38 (0.98–1.94)	0.070
Hematological malignancies	3.25 (2.70–3.91)	<0.001	2.07 (1.54–2.79)	<0.001
Chronic lung disease	1.68 (1.37–2.06)	<0.001	1.49 (1.07–2.06)	0.017
Chronic liver disease	0.73 (0.58–0.92)	0.007	0.98 (0.68–1.48)	0.869
Chronic heart disease	1.16 (1.00–1.34)	0.052	1.13 (0.89–1.43)	0.313
Respiratory distress	2.89 (2.46–3.39)	<0.001	1.76 (1.19–2.60)	<0.001
Sepsis/septic shock	0.64 (0.52–0.79)	0.001	0.97 (0.62–1.50)	0.887
Hypovolemic shock	0.30 (0.18–0.48)	<0.001	0.53 (0.22–1.29)	0.162
Metabolic acidosis	0.69 (0.51–0.94)	0.018	0.88 (0.98–1.38)	0.382
provider worried	0.79 (1.00–6.47)	0.830	0.97 (0.95–1.06)	0.682
MEWS	1.15 (1.01–1.19)	<0.001	0.99 (0.93–1.04)	0.571
SpO_2_/FiO_2_ ratio	0.98 (0.98–0.99)	<0.001	0.97 (0.95–0.98)	<0.001
Noninvasive mechanical ventilation	1.97 (1.32–2.93)	0.001	1.52 (1.07–2.52)	<0.001
High-flow nasal cannula	9.61 (7.89–11.71)	<0.001	4.52 (3.56–5.74)	<0.001
Suctioning of the airway	8.31 (6.79–10.17)	<0.001	4.63 (3.59–5.98)	<0.001
Vasopressor support	2.34 (1.84–3.07)	<0.001	1.76 (1.22–2.54)	0.002

OR = odds ratio; 95% CI = 95% confidence interval; MEWS = Modified Early Warning Score; Hosmer and Lemeshow test *p* = 0.116, Nagelkerke *R*^2^ = 0.307.

**Table 5 jcm-11-01736-t005:** Univariate and multivariate logistic regression analyses for repeat medical emergency team activation for respiratory causes.

Variables	Univariate		Multivariate	
	OR (95% CI)	*p*	OR (95% CI)	*p*
Age	1.00 (0.99–1.01)	0.704	1.00 (0.99–1.00)	0.263
Male (reference: female)	1.17 (0.96–1.44))	0.112	1.19 (0.93–1.51)	0.172
Hematological malignancies	2.70 (2.11–3.46)	<0.001	2.65 (1.94–3.62)	<0.001
Chronic lung disease	1.37 (1.08–1.72)	0.008	1.32 (1.00–1.76)	0.055
Chronic heart disease	1.14 (0.95–1.38)	0.166	1.22 (0.96–1.56)	0.109
MEWS	1.11 (1.06–1.17)	<0.001	1.01 (0.95–1.07)	0.755
SpO_2_/FiO_2_ ratio	0.98 (0.98–0.99)	<0.001	0.98 (0.98–0.99)	<0.001
High-flow nasal cannula	5.14 (4.10–6.45)	<0.001	2.86 (2.22–3.72)	<0.001
Noninvasive mechanical ventilation	1.04 (0.65–1.67)	0.862	1.09 (0.62–1.91)	0.767
Suctioning of the airway	5.64 (4.42–7.19)	<0.001	4.98 (3.69–6.71)	<0.001

OR = odds ratio, 95% CI = 95% confidence interval; MEWS = Modified Early Warning Score. Hosmer and Lemeshow test *p* = 0.330, Nagelkerke *R*^2^ = 0.265.

## Data Availability

The data presented in this study are available on request from the corresponding author. The data are not publicly available due to privacy and ethical restrictions.

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
