# Peer review of "Predictors of Repeat Medical Emergency Team Activation in Deteriorating Ward Patients: A Retrospective Cohort Study"

_jcm, 2022, doi:10.3390/jcm11061736_

Round 1
Reviewer 1 Report
Thank you for the opportunity to review your work.
Comments:
Abstract: " We used prospectively". It is a retrospective study, so the phrase should be revised.
Material/ methods: Please add the objective of the study. Also, add which procedures/ situations MET is called for.
Table 1: Data for " Overall" could be omitted, making it reader-friendly
The discussion needs to be improved. There is a lot of data that has been shared and not discussed. What does this study add to the literature?
Author Response
Thank you for your comments.
We have made corrections according to your comments.
Please refer to the attached file, thank you.

Reviewer 2 Report
I welcome the study performed by Lee et al. In this retrospective analysis, the authors analyzed factors associated with repeat AMET activation and morbidity and mortality associated with it for a large dataset. This is especially important as only few studies have assessed factors associated with repeat METs activation and mortality associated with it. The manuscript is very well written and answers a key questions. With that in mind, I would like to point out some recommendation to help strengthen the study. Methods
1. My Major concern is the methodology section where the dataset, to the best of my knowledge does not exclude patient who may have been transferred to ICU after 1st MET call. These creates a major limitation in the study.
2. For a fair comparison to answer the clinical question, both group should equal chance of getting the repeat METS call. As patient transferred to ICU will not have a repeat MET call limiting the comparison of those groups. As can be seen in Table 3, were almost 1068 had arterial line (which is an indication for ICU transfer in most places) + NIV + vasopressor therapy + HFNC during the first MET call, suggesting most likely a transfer to ICU.
3. If those procedures were done on the hospital ward and patient were kept on hospital ward after initiation of vasopressor therapy of insertion of arterial line + NIV + HFNC, I will suggest the author to include the information as that is not practice in majority of the hospital across the globe.
4. The author may want to include transfer to ICU in both groups after exclusion within 24 -48 hours of MET as the author mentioned in the discussion “Repeat MET activation for at-risk or deteriorating patients is associated with a high risk of ICU admission” but the outcome include unplanned ICU admission.
5. Another exclusion, will be to exclude patient have repeat MET after 48 hrs, as from my understanding, the hypothesis is missed opportunity to escalate care most likely led to poor outcome. MET separated by 48 hrs is highly unlikely to be from worsening of the primary event where the opportunity was missed.
6. My recommendation to the author is to exclude this group reanalyze the data and resubmit.
Author Response

(The authors gave the same response as above.)

Round 2
Reviewer 2 Report
Thank you for taking my suggestions into consideration and editing the manuscript accordingly. The study appears much better.
Introduction: The introduction appears repetitive and can be shortened.
Please remove
“A medical emergency team (MET) is a group of healthcare providers who identify patients at risk and immedi-34 ately manage deteriorating ward patients” is a repetition of previous statement and can be removed.
Rephrase
Increased MET activation is associated with good clinical outcomes [7]. To “Increase utilization of MET services is associated…
In addition, cut down on how MET is good and focus on why identifying risk factors for repeat MET activation is important as that is essentially the clinical question this study aims to answer.
Material and methods
In previous study [8] reported that the most common repeated MET activation occurred within 48 h of the first MET 58 activation.
Rephrase to “Repeat MET activation usually occurs within 48 hrs of previous MET activation.
In logistic regression model, any particular reason chronic heart disease, metabolic acidosis, provider worried and SOFA were not included as they seem to either be statistically significant or have p value of <0.10. Similarly in terms of intervention NIV has a significant impact. Please do include them in the model in both logistic regression model.
For everywhere you mention mortality shown in your study, please add unadjusted mortality.
Discussion
Page 8, line 264-266.
I will suggest an edit, as saying that such management is possible and showing in-hospital mortality with repeat MET close to 60% contradict each other.
I will suggest “in places with limited resources, patient with high-risk factors may be appropriate with the time-limited trial of these interventions, provided there are adequate resources for close monitoring. Any deterioration should prompt transfer to a higher level of care.”
Another point, previous studies have not shown NIV, HFNC, vasopressor or even frequent airway suctioning to be risk factors of repeat MET call, as those patients are in ICU. Edit the manuscript to add this to your limitation. You may add the majority of healthcare around the globe with limited ICU beds have practice patterns similar to ours with references.
Author Response
We thank you and the reviewers for your thoughtful suggestions and insights. The manuscript has benefited from these insightful suggestions. The manuscript has been rechecked, and the necessary changes have been made in accordance with the reviewers’ suggestions. The responses to all comments have been prepared and attached herewith. Changes made to the text are highlighted in red color. Thank you for your consideration.
